# Towards Physical Understanding in Video Generation: A 3D Point Regularization Approach

**Yunuo Chen**[1,2][*]   **Junli Cao**[1,2],   **Vidit Goel**[2],   **Sergei Korolev**[2],
**Chenfanfu Jiang**[1],   **Jian Ren**[2],   **Sergey Tulyakov**[2],   **Anil Kag**[2]
[1]University of California, Los Angeles,   [2]Snap Inc.

## Abstract

We present a novel video generation framework that integrates 3-dimensional geometry and dynamic awareness. To achieve this, we augment 2D videos with 3D point trajectories and align them in pixel space. The resulting 3D-aware video dataset, **PointVid**, is then used to fine-tune a latent diffusion model, enabling it to track 2D objects with 3D Cartesian coordinates. Building on this, we regularize the shape and motion of objects in the video to eliminate undesired artifacts, *e.g.*, non-physical deformation. Consequently, we enhance the quality of generated RGB videos and alleviate common issues like object morphing, which are prevalent in current video models due to a lack of shape awareness. With our 3D augmentation and regularization, our model is capable of handling contact-rich scenarios such as task-oriented videos, where 3D information is essential for perceiving shape and motion of interacting solids. Our method can be seamlessly integrated into existing video diffusion models to improve their visual plausibility.

## 1   Introduction

The recent development of video diffusion models has attracted significant research interests and seen tremendous progress [5, 35, 15]. With vast amounts of training data and sophisticated architectures, video models have greatly improved in expressiveness and aesthetics [6, 37]. Today, state-of-the-art video models can realistically represent content from the real world with high creativity, enabling various applications in the entertainment industry and scientific research [27]. However, existing video diffusion models primarily focus on improving the appearance of content and the smoothness of motion. The models are trained in a way to understand the movements of 2-dimensional pixels, where the object movements in 3D space is approximated through the changing of RGB values.

As a result, the underlying 3-dimensional shapes and motions, which represent how objects truly exist in the physical world, are not learned well in the existing video generation models. Even worse, we observe that when the video diffusion models neglect the underlying 3D information, they struggle to understand how objects occupy space, change shape and location, and interact with each other, which is essential for generating videos with complex motion. For instance, imaging a situation when perceiving a scene with a human performing specific tasks, *e.g.*, *hands tying shoelaces*, viewpoint projection and occlusion make it impossible to capture the true 3D shape of the hand and the shoelaces using 2D pixels, complicating the modeling of how they make contact. Without understanding the underlying geometry, objects in the video may change shape arbitrarily or even appear and disappear suddenly, leading to the common issue of *object morphing* (as example results from prior arts in Figure 1). This problem is exacerbated for complex objects like humans; without 3D awareness, limbs and body parts may move unnaturally or change shape abruptly, resulting in dislocated arms or twisted bodies.

---

[*]Work done during internship at Snap Inc.

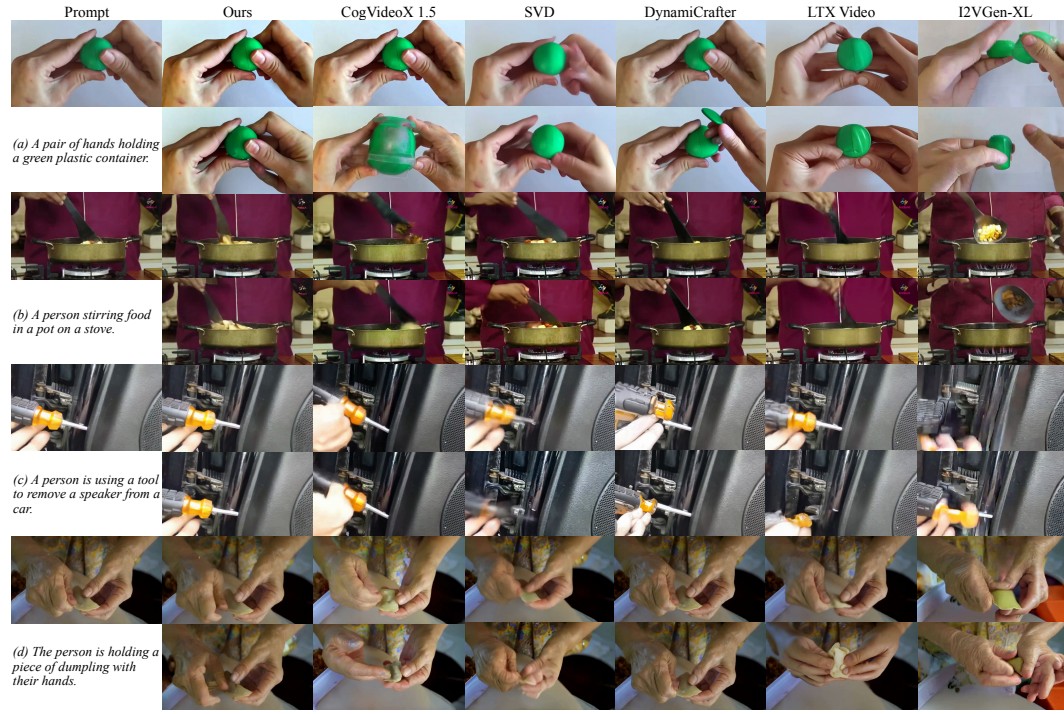

Figure 1: **Comparison on Task-Oriented Videos.** We present videos generated by different baselines (CogVideoX 1.5 [43], SVD [5], DynamiCrafter [42], LTX Video [14] and I2VGen-XL [46]) and compare them with our method. We use the same input conditions for all methods (except that SVD is conditioned only on the image). It can be observed that existing baselines often exhibit severe distortions of hands or objects during human hand-object interactions. In contrast, our method preserves the shapes of both the hand and object during such interactions.

To address the above issue, we aim to improve video diffusion models towards the better understanding of physical world, thus generating more reasonable shape and motion. Previous works have attempted to control shape and motion directly in 2D space [34, 25]. Nonetheless, without a comprehensive understanding of 3D geometry, these methods are limited to directing pixel movement confined within the 2D plane, making them unable to handle out-of-plane motions or fully resolve 3D dynamics [25]. One straightforward approach could be to model the complete 3D geometry of objects using representations such as NeRF [28] or 3D Gaussian-Splatting [20], essentially leading to 4D generation [3, 48]. While these methods provide accurate 3D understanding, they are computationally intensive and generally limited to generating simple scenes with only a few specific types of objects.

To bridge the gap in leveraging 3D information for improved video generation, we propose a method to augment and regularize video diffusion models using 3D point clouds. Specifically, our model enriches 2D videos with out-of-plane data—absent in traditional RGB video models—without requiring full-scale 3D reconstruction. Instead of generating a complete mesh or point cloud for the video diffusion model, we track the movement of 2D pixels in 3D space, creating a pseudo-3D representation of objects in the video. To support this optimization, we develop a 3D-aware video dataset, named **PointVid**, which includes segmented objects in pixel space along with their corresponding 3D dynamic states. We fine-tune video diffusion models on the **PointVid** dataset to extend their generative capabilities to 3D shape and motion, enhancing the model's ability to understand object behavior in the physical world. By leveraging prior knowledge of 3D space, we use this additional information to guide pixel movements toward more visually plausible outcomes.

Incorporating out-of-plane 3D information as a prior, our approach does not rely on specific architectures of the diffusion denoiser and thus can be integrated into various video models. To demonstrate this, we implement our method on two base models with distinct architectures: U-Net-based I2VGen-XL [46] and DiT-based CogVideoX 1.5 [43]. Guided by 3D trajectories, our model achieves superior quality in terms of smooth transitions of shape and motion compared to existing works, generating visually plausible videos. We summarize our contributions as follows:

1. **Injecting 3D Knowledge into Video Diffusion.** We propose a novel *3D point augmentation* strategy to bridge the 3D spatial domain of the physical world with the pixel space of 2D videos. By explicitly incorporating 3D information into the video diffusion process, our method significantly enhances video generation quality, especially in challenging scenarios such as *solid object interactions*.

2. **Regularization with 3D Prior.** We introduce a novel regularization strategy to guide the point cloud diffusion process, ensuring that the generated point cloud is well-structured. This structured point cloud, in turn, enhances the RGB video generation by providing robust 3D priors for better spatial coherence and alignment.

3. **3D-Aware Video Dataset.** We curate a 3D-aware video dataset, **PointVid**, by tracking 3D points in the first frame of a given video. We segment the first frame to obtain foreground objects and mainly track these points in the original video. The resulting dataset contains 3D coordinates corresponding to the first frame, pixel aligned with the video dataset.

## 2   Related Work

**Video Diffusion Models.** Following the success of diffusion-based text-to-image (T2I) models [1, 30, 11, 24, 22, 26, 38, 23, 12, 19], many video generation models have been proposed in the literature [5, 35, 27, 15]. These models allow video generation following input conditioning such as text and image. Video diffusion models can be categorized into three categories. First, auto-regressive models like VideoPoet [21] follow the causal language model paradigm of next token prediction given previous tokens. These models encode the input conditioning into latent tokens using tokenizers such as MAGVIT [44].

Second, pixel-space diffusion models [27, 15, 35] directly model the video synthesis on the pixel space to avoid artifacts arising in the compressed VAE latent spaces. While these models provide more photo-realism and better motion dynamics, they typically only offer low-resolution video generation due to a higher computational cost in generating high-resolution videos. These base models are augmented with super-resolution components to increase the video resolution resulting in deep cascaded diffusion models. Imagen-video [15], a UNet-based architecture, generates videos at a low-resolution and utilizes a sequence super-resolution models to increase the video resolution. Make-a-video [35] uses a T2I as prior to encode text input and trains a deep cascade of spatial and temporal layers to generate high resolution videos. SnapVideo [27] replaces the popular UNet architecture design with a FIT network for improved generation efficiency.

Third, latent diffusion models transform the raw pixel space into a compressed spatio-temporal latent representation. It enables training a higher resolution and higher capacity base model instead of pixel-space diffusion models. These include Stable Video Diffusion [5], SORA [6], VideoCrafter2 [8], ModelScopeT2V [39], Mochi [37], LTX Video [14], CogVideoX [43, 16] etc. While latent models are typically single-stage pipelines, they can extend to deep cascade pipelines to offer even higher-quality video generations. MovieGen [31] proposed a transformer-based cascade latent diffusion model.

While our approach is applicable to various types of video diffusion models, we select two models with distinct underlying architectures— I2VGen-XL [46] and CogVideoX-1.5 [43]—as our base models, and demonstrate how incorporating 3D geometry and dynamics improves video generation.

**Dynamics Aware Video Generation.** There have been various works [34, 25, 47] that instill physically plausible and temporally consistent dynamics in video generation. PhysGen [25] added simple control such as force or torque to an object in within the image, enabling simple physically plausible dynamics to the resulting video. MotionCraft [34] synthesized videos by incorporating optical flow in the noise diffusion process. This allows image animation with a temporally consistent video content. Physdreamer [47] tried distilling dynamic priors of static 3D objects from the video generation models. It creates a physical material field around the 3D object, thereby easily synthesizing 3D dynamics under arbitrary forces. Although these works are able to incorporate simple dynamics to a static image, they assume that the underlying diffusion model understands and captures the object dynamics. This assumption is not necessarily true, as pointed by VideoPhy [4], current video generation models suffer from many inconsistencies, both physically implausible as well as temporally inconsistent. There have been other works along the line of incorporating camera intrinsics during video generation. 3D-Aware Video-Generation [2], first trains a model by jointing diffusing noise along a motion and content codes while sampling different camera poses. The model

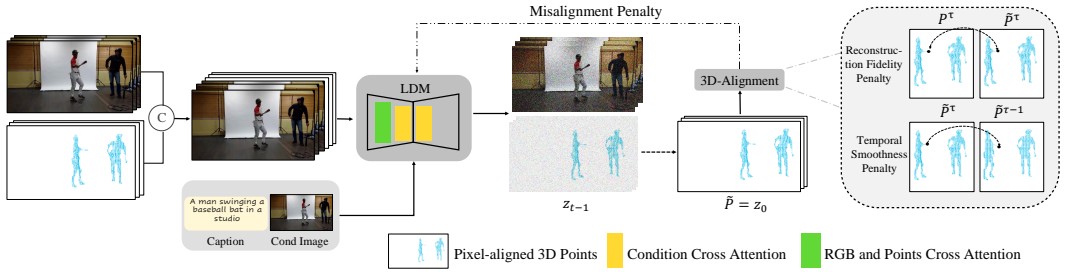

Figure 2: **Method Overview.** During training, we sample video-point pairs, concatenate them along the channel dimensions, and use the augmented data to train a latent diffusion model. We introduce cross-attention between video and point data in corresponding channels to enhance alignment between the two modalities. The model predicts both RGB video and 3D points, leveraging the 3D information to further regularize video generation by applying a misalignment penalty during the diffusion process. During inference, the model generates both video and points from random noise, conditioned on a text-image prompt.

is trained using a generative adversarial network based discriminator which discriminates the resulting video, by difference between frames and time duration.

In contrast to previous works, we incorporate 3D knowledge by jointly diffusing point clouds alongside the video modality. This enhances the diffusion model's understanding of 3D world during training and improves the visual plausibility and spatio-temporal consistency of the generated videos.

## 3 Method

As mentioned earlier, current video models primarily learn moving objects as sequences of 2D pixels, without considering their true 3D states. Their understanding of the physical world is limited to a projected plane, making it challenging to resolve changes in shape and motion that are not visible in the projected view. To address this, we propose incorporating a partial 3D representation into the generation pipeline. The inclusion of 3D information as a second modality will not only enhance the capability of the video model but also guide the generation of RGB videos through joint optimization.

In this section, we outline the details of our 3D-aware video generation pipeline, which consists of three essential components: First, we introduce our novel 3D-aware video dataset, **PointVid**, in subsection 3.1, providing videos with spatial information orthogonal to color dimensions. Next, in subsection 3.2, we present our joint training approach, enhancing the video model's ability to incorporate 3D geometry and dynamics. Finally, we describe our 3D regularization strategy in subsection 3.3, designed to resolve non-physical artifacts in the generated results using the newly incorporated modality.

### 3.1 PointVid Dataset

We build our 3D-aware **PointVid** dataset upon standard 3-channel videos of shape $[T, H, W, C]$, where $C = 3$ encodes the RGB channels, height $H$ and width $W$ digitize 2D images into pixels, and $T$ represents the temporal dimension that sequences the pixel data over time.

The key challenge lies in selecting a 3D representation that is suitable for learning by a video model. Explicit 3D representations, such as meshes (surface or volumetric), point clouds, or voxels, can effectively model geometry but often have variable sizes due to irregular object shapes, complicating the diffusion training process. Moreover, aligning 3D points with 2D pixels in video remains a non-trivial task. In contrast, implicit representations like NeRF [28] typically require substantial time to convert into an explicit form for direct usage, making them impractical for training purposes.

Recent advancements in 3D point tracking [41] have made this task more feasible. Instead of representing moving objects as sequences of meshes or point clouds with arbitrary resolution, we focus exclusively on tracking the movements of pixels. Here, 2D pixels are unprojected into 3D space and tracked throughout the video. This allows the 3D locations of the pixels to be treated as three

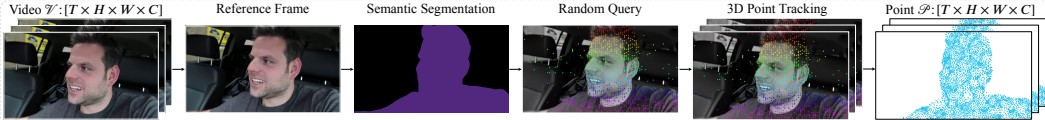

Figure 3: **PointVid Dataset Generation**. Given an input video, we use the first frame as a reference frame and perform semantic segmentation to obtain masks for foreground objects. Next, we randomly sample pixels with a distribution favoring pixels inside foreground objects. We perform 3D point tracking on these queried pixels, and map these points to the input video frames. The resulting data point contains 3D coordinates of tracked foreground pixels while remaining pixels are zeroed out.

additional attributes, similar to color intensities. As a result, we can format the 3 spatial coordinates as extra channels, yielding a point tracking tensor $\mathcal{P}$ of shape $[T, H, W, C]$, matching the dimensions of the RGB video. In practice, we store the 3D tracking information using 2D pixel coordinates $(u, v)$ and depth $d$, requiring only a simple camera unprojection to reconstruct the 3D world coordinates $(x, y, z)$. By using $(u, v, d)$, the first two channels correspond to the pixel's position projected onto the 2D plane, which aligns precisely with the moving point in the RGB video, thus simplifying the task of learning the alignment between video and point tracking.

We visualize the workflow of generating point tracking in Figure 3. For each RGB video $\mathcal{V}$, we extract the first frame as a reference image and perform semantic segmentation using Grounded-SAM-2 [33, 32], yielding masks for the foreground objects. Next, we randomly sample pixels on the reference image with a main focus on those inside the foreground objects. These queried pixels are then tracked with trajectories of 3D Cartesian coordinates $(x, y, z)$ throughout the video using SpaTracker [41]. Finally, we post-process the tracked results to align the shape with the video: we fill out the pixels in the foreground mask to store the pixel's spatial positions, formatted as 3 channels, and discard pixels in the background (setting all background pixels to zero). Since we apply sparse queries for tracking efficiency, some foreground pixels may not have a corresponding trajectory; we use KDTree searching to interpolate the positions for such pixels.

One remaining issue we encounter in the tracked results is that they sometimes contain high-frequency noise, possibly due to the imprecision of the tracking. This causes problems when the video model attempts to learn from its distribution, as the backbone diffusion model relies on sampling with random noise. To address this, we apply a Kalman Filter for 3D point tracking to remove temporal high-frequency noise and recover its primary motion.

## 3.2   Point Augmentation in Diffusion Denoiser

With our 3D-aware video dataset, we now aim to extend the capabilities of the video model to generate not only the RGB video $\mathcal{V}$ but also the corresponding 3D representation $\mathcal{P}$. Aligning the video with the point cloud enables the model to inherently acquire 3D knowledge. The augmented video-point model learns not only how the object appears in a 2D color image but also its 3D geometry.

Our augmentation pipeline is illustrated in Figure 2. It is worth noting that training the video-point model entirely from scratch is unnecessary, as training video diffusion models already requires significant computational resources and large-scale datasets. Instead, it is more practical to fine-tune a pretrained video model to incorporate the point-tracking modality. Additionally, our injection of 3D points is agnostic to the network architecture of the pretrained video model. In this work, we employ both UNet- and DiT-based denoisers—specifically, I2VGen-XL [46] and CogVideoX-1.5 [43]—both conditioned on text-image prompts. While our training pipeline incorporates an additional point modality, inference remains conditioned only on text-image pairs, consistent with the base model.

To collectively learn from and generate $\mathcal{V}$ and $\mathcal{P}$, the straightforward way would be to feed them together to the diffusion model. Since we are matching the shapes of video and point tensors, we can concatenate them along the channel dimension, resulting in an point-augmented video $\mathcal{VP}$ with a shape of $[T, H, W, 2C]$. This concatenated video can be seamlessly integrated into the denoiser by simply adjusting the input/output dimensions to accommodate the additional channels. This increase in dimensionality does not overshadow the weights of the original model; instead, only new projection layers are introduced to handle the added modality. To leverage pretrained RGB weights, we can load the pretrained model for the corresponding RGB channels and initialize the remaining weights as

zeros. For the conditioning frame, we continue using the first RGB frame and repeat the condition image tensor to prompt the generation of both video and point tracking, as our goal is for the model not to rely on the initial point positions during the inference stage. This strategy is effective because the same reference frame is used for generating the point-tracking data earlier.

In addition to the pixel-wise alignment of video and points during dataset construction, we introduce a cross-attention layer at the input of the latent diffusion model. This layer links the first and second halves of the channel dimensions—corresponding to color information and spatial location in the latent space—to further enhance their alignment. We apply two passes of cross-attention, with each modality alternately serving as the query, while the other provides the key and value in each pass.

Incorporating the aforementioned changes, the training pipeline proceeds as usual: we encode $\mathcal{VP}$ into the latent space $z$, add noise to obtain $z_t$ using randomly sampled noise $\epsilon$ and a diffusion step $t$. The denoiser model then predicts the added noise and compares it with the ground-truth noise, where we apply DDIM [36] for diffusion sampling. With this modified pipeline, we fine-tune the diffusion model on **PointVid** dataset to obtain our video-point generation model. At inference, given an image and a text prompt, the model takes in double-sized random noise and predicts both the RGB video and the corresponding 3D trajectories.

### 3.3 Point Regularization

With the model capable of generating pixel-aligned video and points, we leverage this additional modality to enhance RGB video generation. To achieve this, we introduce regularization terms on the point tracking component of the diffusion output, aligning it with the ground truth, which in turn implicitly improves the RGB video component (see Figure 2).

Firstly, we need to recover directly usable point trackings for evaluation in 3D space. Since injecting random noise into the point cloud would likely render it unusable, we employ diffusion sampling to recover a noise-free stage before applying regularization. At each training step, given a latent input $z$ and its corresponding noisy stage $z_t$, we apply DDIM denoising to iteratively recover $z_0$. Following the approach of [45], we adopt gradient checkpointing to freeze gradient flow except during the final step, thereby preventing memory overflow. The denoised latent $z_0$ is subsequently decoded into explicit 3D point tracking $\tilde{\mathcal{P}}$, representing the sampled 3D trajectory of the video.

With the explicit modeling of trajectories as a 3D representation, we introduce regularization terms to improve the quality of the generated results, focusing on motion fidelity and temporal smoothness. For simplicity, let us assume $\tilde{\mathcal{P}}$ is reshaped to $T \times N \times 3$, and use $\tilde{\mathcal{P}}^\tau$ to denote the point cloud of shape $N \times 3$ at time $\tau$.

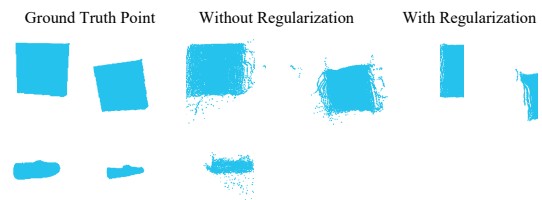

Ground Truth Point    Without Regularization    With Regularization

A common issue that arises during point tracking generation is the presence of excessive noise (see Figure 4 middle). This noise can originate from various sources, such as the variational autoencoder (VAE) used to encode $\mathcal{P}$ into the latent space. High-frequency noise in both spatial and temporal dimensions tends to obscure the main shape and motion, thereby impairing

Figure 4: **Point Regularization.** The reconstructed point cloud in the diffusion output often contains noise and deformations (middle). This issue is mitigated using our point regularization (right). The synthetic point cloud above (*e.g.*, *box* and *shoes* falling on the ground) is generated by Kubric [13] and trained with our pipeline.

the video model's ability to track 3D points effectively. To mitigate noise and increase fidelity of reconstruction, we propose the reconstruction loss $\mathcal{L}_{\text{recon}}$ as follows:

$$\mathcal{L}_{\text{recon}} = c_0 \sum_{\tau=0}^{T} \|\tilde{\mathcal{P}}^\tau - \mathcal{P}^\tau\| + c_1 \sum_{\tau=1}^{T} \|\tilde{\mathcal{P}}^\tau - \tilde{\mathcal{P}}^{\tau-1}\| + c_2 \sum_{\tau=2}^{T} \|\tilde{\mathcal{P}}^\tau - 2\tilde{\mathcal{P}}^{\tau-1} + \tilde{\mathcal{P}}^{\tau-2}\|. \quad (1)$$

Here the first term penalizes the difference between generated point cloud and the noise-free ground truth $\mathcal{P}$, ensuring generation fidelity, while the second term penalizes the difference between consecutive time step to encourage temporal smoothness; the third term further enforeces acceleration smoothness, discouraging abrupt changes in velocity. We selected the weights $c_i$ so that each term are about the same scale initially.

Apart from the microscopic high-frequency noise, we observe a macroscopic shape deformation in many of the generated point cloud (see Figure 4 bottom middle). Such deformation of arbitrary degree is undesirable as most solid objects, including human body, inderently requies conservation of volumn and local rigidity. Violating this can lead to unnatural shape deformation and morphing, which is considerd undesired in our generation task. Hence we propose the rigidity loss $\mathcal{L}_{\text{rigid}}$ to enforce local shape preservation:

$$\mathcal{L}_{\text{rigid}} = \sum_{\tau=1}^{T} \sum_{i} \sum_{j \in \mathcal{N}(i)} \left( \text{dist}(\tilde{\mathcal{P}}_i^\tau, \tilde{\mathcal{P}}_j^\tau) - \text{dist}(\tilde{\mathcal{P}}_i^0, \tilde{\mathcal{P}}_j^0) \right)^2. \tag{2}$$

Here, we apply a kNN search on the reference frame to construct local neighbor pairs, then penalize changes in distance between these pairs.

By combining reconstruction and rigidity loss, we enhance the generation quality of the 3D point cloud, as shown in Figure 4 (right). This improvement in the point space implicitly promotes shape preservation and smooth motion transitions in the RGB video, as our video-point model closely aligns the two modalities. To prevent the geometric regularizations from dominating the semantic aspects, we jointly optimize these regularization terms with the standard diffusion denoiser losses $\mathcal{L}_{\text{diff}}$ [36] during training, resulting in an overall loss of the form (where $\lambda_{[\cdot]}$ are corresponding weights):

$$\mathcal{L} = \lambda_{\text{diff}} \mathcal{L}_{\text{diff}} + \lambda_{\text{recon}} \mathcal{L}_{\text{recon}} + \lambda_{\text{rigid}} \mathcal{L}_{\text{rigid}}. \tag{3}$$

## 4 Experiments

In this section, we demonstrate the effectiveness of our proposed method through both qualitative and quantitative evaluations. Our fine-tuning approach can be seamlessly integrated into different base video diffusion models. In our experiment, we adopt two baseline image-to-video models with distinct architectures: a UNet-based model, I2VGen-XL [46], and a DiT-based model, CogVideoX 1.5 [43]. We refer readers to our supplementary material for more details on the training setup.

### 4.1 Qualitative Results

Although our model is designed to generate sequences of 2D images, our point augmentation and regularization improve its ability to capture inherently 3D motions. To evaluate this, we test our models on unseen images and visualize the generated RGB videos. For generation quality, we primarily assess visual plausibility, with a particular focus on shape consistency and preservation. We refer readers to our supplementary document and video for additional results.

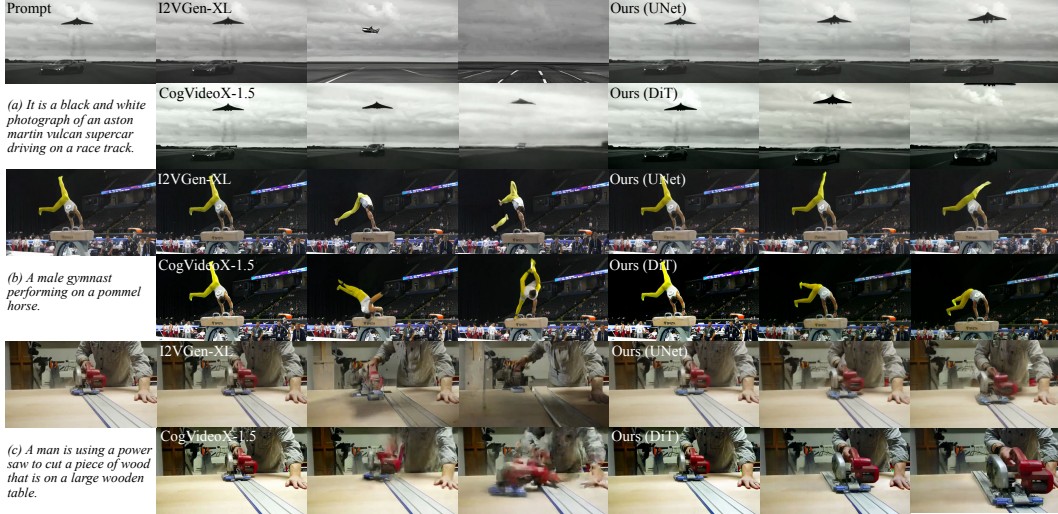

Figure 5: **Qualitative Comparison.** We compare our UNet and DiT models against their respective baselines. The results show that both base models exhibit unrealistic artifacts, such as morphing, while our models ensure smooth transitions in object shape and motion.

Table 1: **Quantitative Evaluation.** We evaluate various aspects of our method against other models using the VBench [17] and VideoPhy [4] benchmarks. The evaluated metrics are as follows: (VBench) SC: subject consistency, BC: background consistency, MS: motion smoothness, AQ: aesthetic quality, IQ: imaging quality; (VideoPhy) PC: physical commonsense. Our UNet and DiT models are compared against their respective baselines, with higher scores highlighted in bold, while the highest achiever among all models is underscored.

| Method | SC ↑ | BC ↑ | MS ↑ | AQ ↑ | IQ ↑ | PC ↑ |
|---|---|---|---|---|---|---|
| SVD | 0.91807 | 0.93563 | 0.97735 | 0.43276 | 0.55174 | 0.34707 |
| LTX Video | 0.89001 | 0.90423 | 0.99152 | 0.44670 | 0.58667 | 0.33400 |
| I2VGen-XL | 0.83704 | 0.89356 | 0.96144 | **0.43140** | 0.60338 | 0.32876 |
| Ours (UNet) | **0.96003** | **0.95120** | **0.98458** | 0.42246 | **0.61003** | **0.37805** |
| CogVideoX 1.5 | 0.91754 | 0.92923 | 0.98798 | 0.44534 | 0.60067 | 0.36549 |
| Ours (DiT) | **0.91768** | **0.92987** | **0.99201** | **0.45298** | **0.60531** | **0.41384** |

**Comparison with base models.** We compare our models side by side with their corresponding base models: I2VGen-XL [46] for the UNet model and CogVideoX 1.5 [43] for the DiT model. Using the same conditioning text and image, we visualize the generated RGB videos in Figure 5. In the videos generated by the base models, noticeable artifacts such as object morphing and unnatural deformations can be observed—for example, the car in (a), the gymnast in (b), and the power saw in (c). In contrast, our models effectively eliminate unnatural shape and appearance distortions, resulting in visually plausible outputs. While our UNet and DiT models differ in architecture and pretrained baselines, we consistently observe a reduction in artifacts compared to their respective base models.

**Task-oriented videos.** A prevalent category of real-world videos features humans performing specific tasks (see Figure 1), such as playing with a toy, cooking food, or handling tools. To highlight the challenges of such scenarios, we evaluate three additional video models: SVD (stable-video-diffusion-img2vid-xt) [5], DynamiCrafter [42], and LTX Video [14], alongside the two baseline models. We observe that these models often struggle to capture localized details, particularly how human hands interact with objects. This challenge stems from the inherently complex 3D nature of these tasks, which cannot be fully captured using only 2D data. As a result, hands and objects frequently undergo severe distortion or blurring. With our 3D regularization, our model effectively preserves the shapes of both hands and objects, ensuring smooth transitions throughout the video and demonstrating consistent quality improvements over the compared models.

## 4.2 Quantitative Results

To quantitatively evaluate the performance of our model, we adopt a test batch of randomly selected images. The test images are sampled from a single batch of video clips (unseen by our model) from the video dataset [9], with the initial frame randomly selected within each clip, totaling 372 images. To ensure consistency, we use the same test batch across all quantitative evaluation metrics (including ablation studies in subsection 4.3 and user studies in subsection A.5). In this section, we compare various aspects of video quality between our two models and their respective baseline models, along with two additional models for reference. The results are summarized in Table 1.

First, we assess the general aspects of video generation quality using VBench [17], evaluating both temporal quality (subject/background consistency and motion smoothness) and frame-wise quality (aesthetic and imaging quality) of the videos. The results show that our fine-tuned models outperform their baselines in most categories, demonstrating the effectiveness of our approach in improving video generation qualities, particularly in content consistency.

Beyond these general aspects, we are particularly interested in how the injection of 3D awareness enhances alignment with real-world physical principles. It is important to note that determining whether a 2D video is physically accurate is largely subjective, as ground-truth 3D physical states are not available. Therefore, we utilize VideoPhy [4], which employs human annotations to train a discriminator that provides a *physical commonsense* score. VideoPhy defines physical commonsense from various perspectives, such as solid mechanics, where solid objects are expected to retain their shape and size throughout the video, aligning with our objective. As shown in Table 1, our models

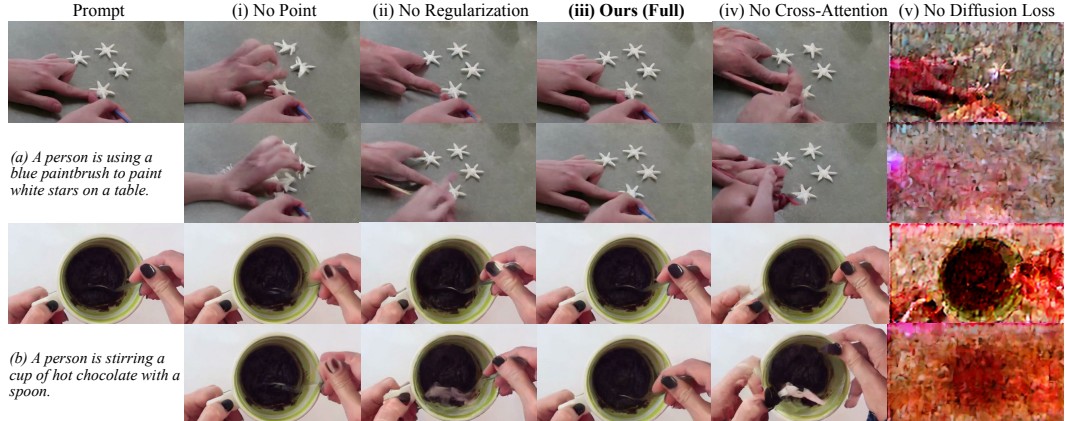

| Prompt | (i) No Point | (ii) No Regularization | **(iii) Ours (Full)** | (iv) No Cross-Attention | (v) No Diffusion Loss |

*(a) A person is using a blue paintbrush to paint white stars on a table.*

*(b) A person is stirring a cup of hot chocolate with a spoon.*

Figure 6: **Ablation Studies on Different Components.** We compare the results from (iii) our full generation pipeline with (i) training with video only, (ii) training with point augmentation only, (iv) training without channel cross-attention, and (v) training without diffusion loss.

achieve a higher score for alignment with physical principles, outperforming their respective baselines and other tested models.

To further demonstrate the efficacy of our 3D point augmentation and regularization modules, we evaluate the mean squared error (MSE) of the predicted 3D point trajectories during diffusion (using the input point tracking tensor as ground truth), as summarized in Table 2. We observe that, compared to an untrained diffusion model, our network design effectively predicts 3D point

Table 2: **Evaluation of Points.** We report the MSE of predicted 3D points for the untrained model, the model without regularization, and the model with regularization, respectively.

| Item | Untrained | No Reg | With Reg |
| --- | --- | --- | --- |
| MSE ↓ | 0.2125 | 0.0031 | **0.0020** |

trajectories after the augmentation stage and further reduces the error through the regularization stage.

## 4.3 Ablation Study

We conduct ablation studies on our UNet-based model to evaluate the effectiveness of our training pipeline design. We visualize the inference results in Figure 6 and use VBench [17] to evaluate the models' subject and background consistency in Table 3. Our two-stage design of point augmentation and regularization progressively enhances the video model.

Table 3: **Quantitative Ablation Studies.** We use VBench [17] to evaluate the subject and background consistency (SC/BC) of our full model, as well as models trained without point augmentation (PA), without point regularization (PR), cross-attention (CS), and diffusion loss (DL).

| Metric | Full | No PA | No PR | No CS | No DL |
| --- | --- | --- | --- | --- | --- |
| SC ↑ | **0.9681** | 0.9434 | 0.9544 | 0.9332 | 0.7166 |
| BC ↑ | **0.9639** | 0.9492 | 0.9522 | 0.9329 | 0.9254 |

Compared to training the model with RGB videos only (Figure 6 (i)), our point augmentation alone (Figure 6 (ii)) injects 3D awareness into the model and improves its ability to perceive 3D shapes, as evidenced by more consistent object shapes in the videos. Our point regularization (Figure 6 (iii)) further improves quality by optimizing point generation and implicitly guiding RGB generation towards higher fidelity. Our channel-wise cross-attention mechanism ensures cross-dimensional and cross-modality alignment, without which video quality may degrade (Figure 6 (iv)). Furthermore, in the regularization stage, we adopt a joint training strategy that combines regularization loss with diffusion loss. We observe that dropping diffusion loss (Figure 6 (v)) causes 3D information to dominate semantics, resulting in completely degraded results.

## 5 Conclusion and Discussion

To summarize, we propose a generation framework that incorporates 3D-awareness into video diffusion models. By tracking objects in 3D and aligning them in pixel space, we elevate traditional video datasets to a new dimension, revealing out-of-plane information previously inaccessible to video models. Through joint training, the video model learns to perceive 3D shape and motion, acquiring physical commonsense that is inherent in 3D. We then apply regularization to refine the

generation process and further enhance the results, eliminating artifacts such as object morphing. Our method can serve as an augmentation for various video models, and we showcase its effectiveness by fine-tuning two widely-used video generation models (I2VGen-XL and CogVideoX 1.5). The results demonstrate a strong capability in reducing non-physical artifacts in video generation, particularly when applied to contact-rich, task-oriented videos.

**Motion Magnitude and Smoothness Trade-off.** In the design of our reconstruction loss (Equation 1), we align the generated 3D points with the ground truth positions while enhancing temporal smoothness by minimizing velocity and acceleration. While we acknowledge that this loss introduces a trade-off between motion magnitude and smoothness, the global dynamics are primarily governed by the position term (to which we assign larger weights). As a result, we observe faithful reconstruction of ground-truth dynamics during training (see Table 2). Moreover, discouraging abrupt changes in velocity is often beneficial, as they typically induce morphing artifacts in generated video (see Figure 5). We refer the reader to subsection A.2 for more details on the training setup.

**Limitations.** We acknowledge a few limitations of our framework. The degree of 3D-awareness, rooted in our video-point joint training, is constrained by the resolution of our 3D points. While our method does not require highly accurate point tracking for 3D prior, objects that are not sufficiently represented may lead to suboptimal 3D perception by the video model. Recent advancements in 3D point tracking [29, 40] could be deployed to enhance the 3D perception in our model. Additionally, the generation quality of our model is limited by the base diffusion backbone and could potentially be improved by employing larger-scale models. We leave these aspects as directions for future exploration.

**Concurrent Works.** Concurrently, several studies have emerged to enhance shape and motion consistency in video generation. [7] uses optical flow as a representation of object motion and does not explicitly model object geometry. [18] proposes using point tracking to improve coherence; however, it only utilizes 2D pixel correspondence without 3D spatial information. On the contrary, our method explicitly models the 3-dimensional geometry and motion to provide spatial guidance to the diffusion model.

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

# A   Technical Appendices and Supplementary Material

Supplementary to the main paper, we provide more details on dataset creation in subsection A.1, more details on implementation in subsection A.2, additional qualitative results in subsection A.3, and a human preference evaluation in subsection A.5.

## A.1   PointVid Dataset Generation

In this section, we present additional details on generating our 3D-aware video dataset, **PointVid**. A visualization of the generation pipeline is provided in the main paper. Our pipeline requires no additional annotations beyond the video-text pairs in common video datasets.

**Preparation**   Starting with any video-text pair, we trim the videos to a fixed length $T$ and shape $H \times W$, extracting the first frame of the clipped video as the *reference frame*. We then apply semantic segmentation to the reference frame using Grounded-SAM2 [33, 32] to automatically generate masks for foreground objects. To streamline the process, we utilize a language model, LMDeploy [10], to infer the main moving objects in the scene based on the video's caption. For example, the caption *A man swinging a baseball bat in a studio* results in *{man, baseball bat}*. This object-level label is subsequently fed to Grounded-SAM2 for prompt-based segmentation.

**Point Tracking**   Having obtained a reference frame and its segmentation, we apply SpaTracker [41] to track pixel movements accordingly. We use the same reference frame to query points during tracking so that the 3D positions align with the first frame's pixel locations initially. We set *downsample* to 1 and *gridsize* to 80 in [41] to enable sparse tracking. We observe that sparse tracking at this resolution provides a sufficient number of pixels while remaining relatively efficient in terms of time. Increasing the resolution or using dense tracking could potentially enhance the results, but we leave this for future exploration. The model outputs point tracking data of shape $T \times N \times 3$, where $N$ is the number of sampled points.

---

**Algorithm 1** Post-processing for Point Tracking

---

**Require:** Tracked points $\mathbf{P} \in \mathbb{R}^{T \times N \times 3}$, Foreground mask $\mathbf{M} \in \{0,1\}^{H \times W}$, Resolution $(H, W)$
**Ensure:** Processed tensor $\mathcal{P} \in \mathbb{R}^{T \times H \times W \times 3}$

1: **Initialize:** $\mathcal{P} \leftarrow 0$             ▷ Initialize all values to zero
2: $\mathbf{P} \leftarrow \text{KalmanFilter3D}(\mathbf{P})$            ▷ Reduce temporoal noise
3: $\mathbf{P}_0 \leftarrow \mathbf{P}[0, :, 0:2]$            ▷ Initial $uv$ of 2D pixels
4: $\mathbf{P}_1 \leftarrow \mathbf{P}_0 \bigcap \mathbf{M}$            ▷ Filter foreground pixels
5: $\mathbf{Tr} \leftarrow \text{KDTree}(\mathbf{P}_1)$            ▷ Use initial $uv$ for searching
6: **for** each foreground pixel $M[u, v] = 1$ **do**
7:      **if** found $\mathbf{P}_1[j] = (u, v)$ **then**            ▷ Tracked pixel
8:          $\mathcal{P}[:, u, v, :] \leftarrow \mathbf{P}[:, j, :]$
9:      **else**            ▷ Untracked pixel
10:          $\text{ind} \leftarrow \text{Query}(\mathbf{Tr}, (u, v), k = 3)$
11:          $\mathcal{P}[:, u, v, :] \leftarrow \text{Interp}(\mathbf{P}[:, \text{ind}, :])$
12:      **end if**
13: **end for**
14: **return** $\mathcal{P}$

---

**Post-processing**   To mitigate temporal noise caused by inaccurate tracking, we first apply a Kalman Filter to the 3D point cloud to estimate a global velocity and eliminate high-frequency noise. The result is then reshaped to $T \times H \times W \times 3$, where the spatial coordinates of pixels are treated as three channels, similar to RGB. To address the difference in shape between the $N$ tracked points and the video resolution, we apply the following rules: if a tracked pixel is not within the foreground mask, we discard its value (effectively setting all background pixels to zero). Otherwise, we assign its value to the corresponding pixel entry in the tensor. Since we use sparse tracking, only a subset of foreground pixels have values. We then apply KDTree searching on the sparse pixels based on their $uv$ coordinates in the first frame. For all untracked pixels within the foreground mask, we approximate their trajectory by interpolating their three nearest tracked pixels. The pseudocode for post-processing is summarized in Algorithm 1.

## A.2    Implementation Details

In this section, we provide more details on our experiments.

**Dataset**    We collect approximately 100K videos from [9] and our internal dataset (each accounting for half) to form our **PointVid** dataset. The videos are processed to a resolution of $640 \times 360$ and cropped to a 2-second length (the exact number of frames may vary due to varying input frame rates, typically it is around 50 frames or more), and then paired with 3D point tracking using [41].

**Model Training**    We implement our pipeline on two baseline models: UNet-based model I2VGen-XL [46] and DiT-based model CogVideoX 1.5-5B [43]. For our UNet model, we use a resolution of $448 \times 256$ with 16 frames; for our DiT model, we use a resolution of $1360 \times 768$ with 48 frames. During finetuning, we train on 8 NVIDIA A100 80G GPUs with a batch size of 4 for UNet model and a batch size of 1 for DiT model.

**Joint Optimization**    To effectively inject 3D-awareness into the video model without degrading the RGB modality (see subsection 4.3 for details), we adopt a joint optimization strategy combining diffusion and regularization losses. In our experiments, we observe that when the regularization loss significantly outweighs the diffusion loss (by orders of magnitude), it tends to suppress motion. Conversely, when the diffusion loss dominates training, non-physical artifacts reappear. In practice, we balance the two losses to maintain comparable magnitudes, which consistently reduces reconstruction misalignment and noticeably mitigates non-physical artifacts. To further preserve the motion magnitude, we apply the regularization loss less frequently (once every $k$ iterations, we use $k = 5$). This adjustment preserves larger motion dynamics at the cost of requiring more training steps.

**Computation Cost**    For our UNet model trained on an A100 GPU with a batch size of 4, we report the additional computational cost as follows: incorporating 3D components and regularization losses increases training time ($\sim$9s per iteration, compared to $\sim$4.75s when fine-tuning on RGB video only) while maintaining comparable GPU memory usage (adding less than 2GB on top of $\sim$45GB).

## A.3    Additional Qualitative Results

**Additional Ablation Results**    We provide additional qualitative results from our ablation studies in Figure 7. Our point augmentation and regularization approach progressively enhance the visual quality of the generated videos, while the channel cross-attention mechanism and diffusion loss are also critical for ensuring high-quality results.

**Additional Qualitative Comparisons**    We provide additional qualitative evaluations of our model compared to other models: I2VGen-XL [46], SVD [5], DynamiCrafter [42]. These include both *task-oriented* videos (see Figure 8) and videos from non-specific categories (see Figure 9), such as humans, animals, and general objects. Our results demonstrate improved shape and motion consistency of objects, effectively minimizing common non-physical artifacts like object morphing.

## A.4    Additional Quantitative Results

While we use a fixed test set in all quantitative evaluations to ensure consistency, we also report the VBench [17] results of our UNet model using the official batch split provided by VBench. The results are summarized in Table 4. We observe similar outcomes as in the main comparison table (Table 1).

Table 4: **Quantitative Results on VBench Test Batch.** Our model outperforms the base model in most evaluation metrics.

| Method | SC ↑ | BC ↑ | MS ↑ | AQ ↑ | IQ ↑ |
|---|---|---|---|---|---|
| Baseline | 0.89703 | 0.94009 | 0.96993 | **0.52809** | 0.62961 |
| Ours | **0.96418** | **0.96722** | **0.98319** | 0.51754 | **0.65428** |

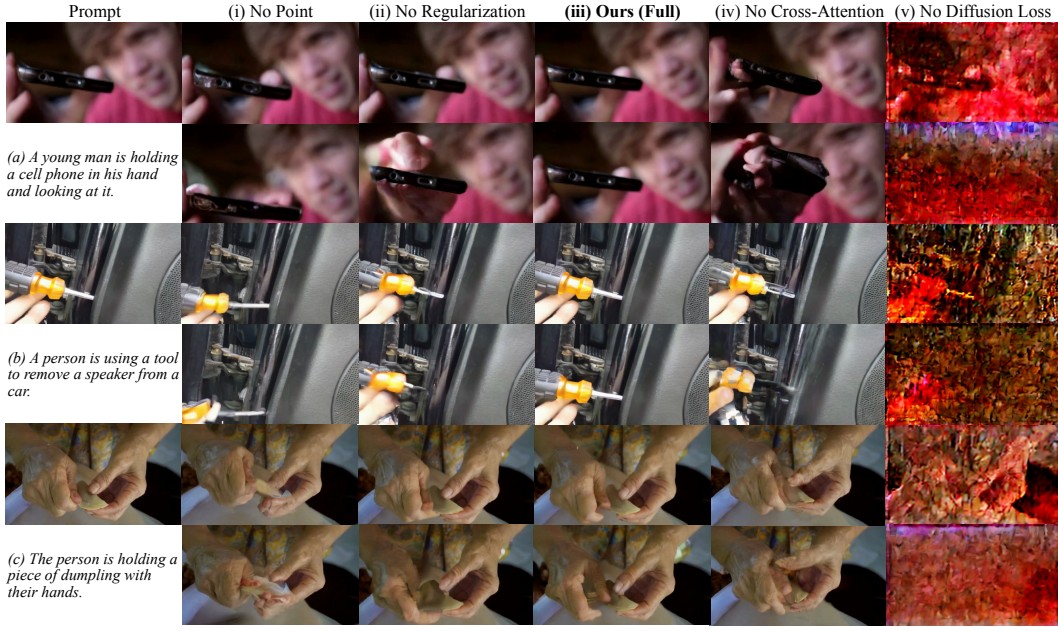

| Prompt | (i) No Point | (ii) No Regularization | **(iii) Ours (Full)** | (iv) No Cross-Attention | (v) No Diffusion Loss |

Figure 7: **Additional Ablation Results.**

## A.5 User Study

Apart from the quantitative evaluation using VBench [17] and VideoPhy [4], we additionally conduct a user study on our UNet model in comparison to its base model I2VGen-XL [46] to assess human preference on the models, focusing on identifying non-physical artifacts and evaluating overall visual plausibility. The following 5 questions are designed for evaluation:

1. Which video appears to have more physically realistic object movements and interactions? (Consider aspects like gravity, collisions, and the natural flow of objects.)

2. (Negative) Which video contains more noticeable **non-physical artifacts**, such as object morphing, stretching, or sudden changes in shape?

3. In which video do the objects maintain a more consistent size, shape, and appearance throughout the entire sequence?

4. Which video better adheres to natural physics laws, such as consistent lighting, shadow behavior, and material properties (e.g., rigidity or fluidity)?

5. Overall, which video feels more coherent and believable based on the physical interactions and behavior of objects?

Given the evaluation questions, we ask 10 labelers to compare our generated videos side by side with results from the base model and select the one that best fits each question. We use the same testing batch used in quantitative evaluation. As shown in Table 5, our method demonstrates significant improvement in terms of physical plausibility by incorporating 3D awareness into the video.

Table 5: **User Study Results.** Our model demonstrates a significant improvement in physical plausibility, as assessed by human labelers. Here, Q2 asks the user to identify negative artifacts in the videos, while the other four questions positively assess physical plausibility.

| Method | Q1 ↑ | Q2 ↓ | Q3 ↑ | Q4 ↑ | Q5 ↑ |
|---|---|---|---|---|---|
| Baseline | 0.138 | 0.862 | 0.135 | 0.132 | 0.137 |
| Ours | **0.862** | **0.137** | **0.865** | **0.868** | **0.863** |

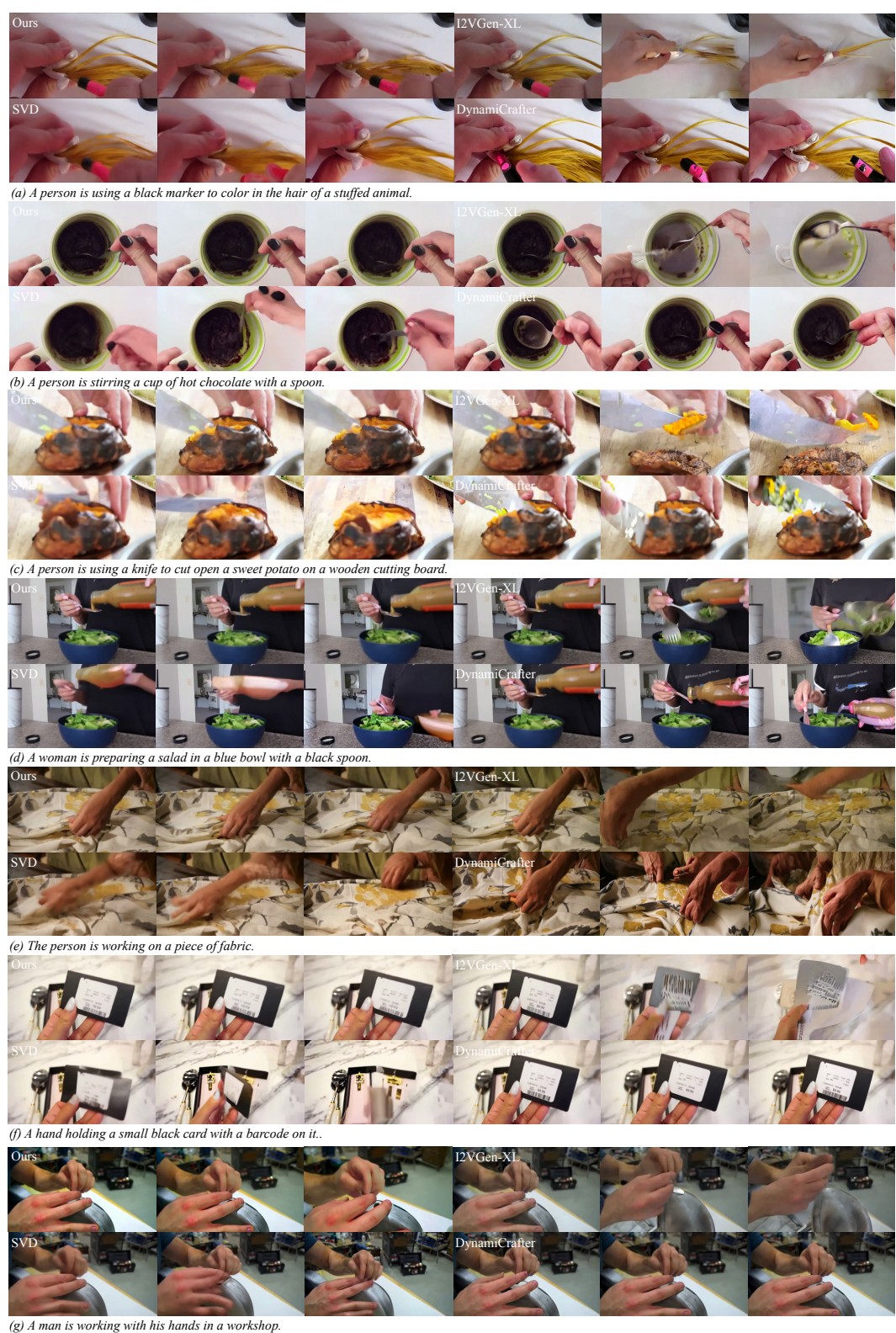

(a) A person is using a black marker to color in the hair of a stuffed animal.

(b) A person is stirring a cup of hot chocolate with a spoon.

(c) A person is using a knife to cut open a sweet potato on a wooden cutting board.

(d) A woman is preparing a salad in a blue bowl with a black spoon.

(e) The person is working on a piece of fabric.

(f) A hand holding a small black card with a barcode on it..

(g) A man is working with his hands in a workshop.

Figure 8: **Additioanl Comparison on Task-oriented Videos.**

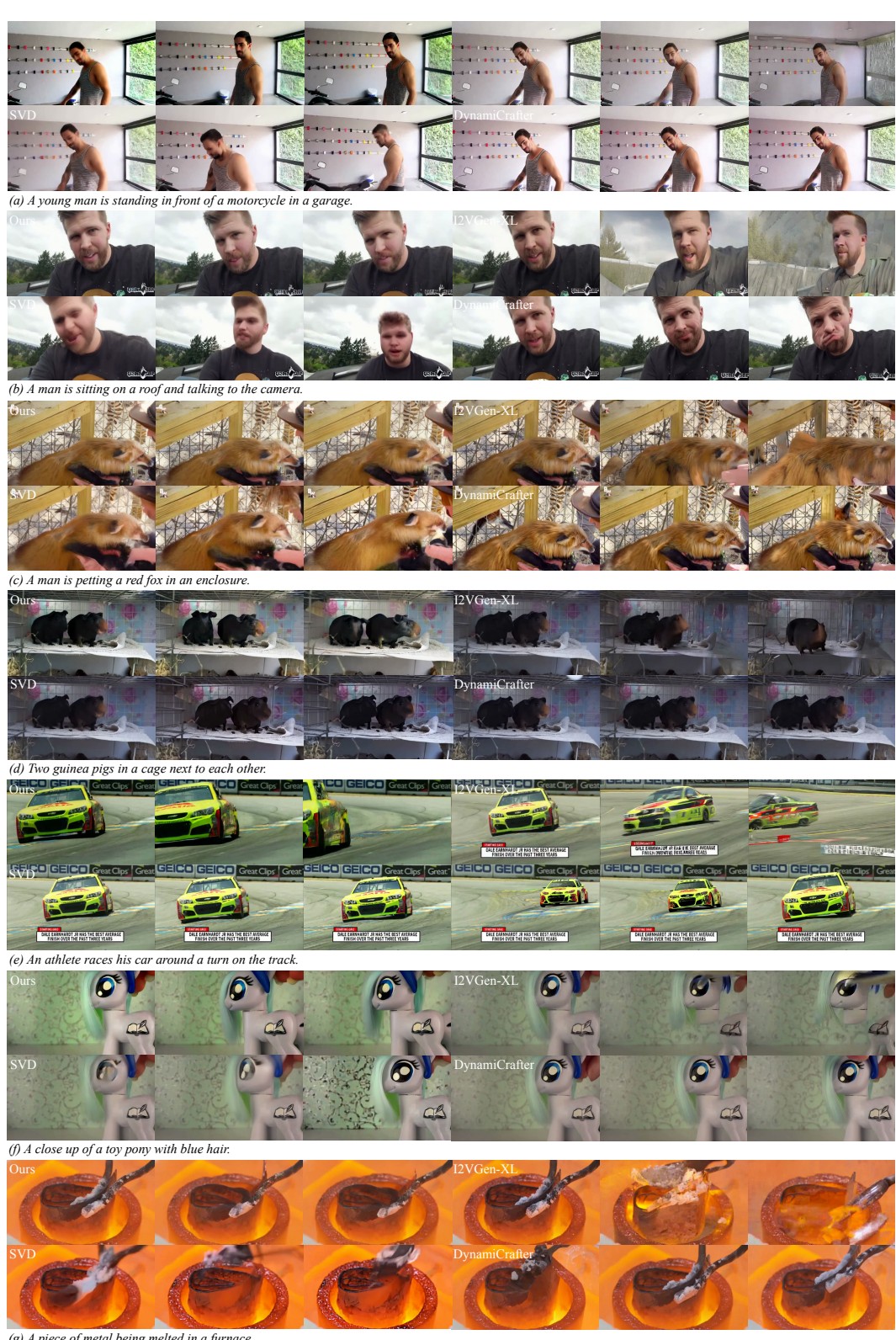

*(a) A young man is standing in front of a motorcycle in a garage.*

*(b) A man is sitting on a roof and talking to the camera.*

*(c) A man is petting a red fox in an enclosure.*

*(d) Two guinea pigs in a cage next to each other.*

*(e) An athlete races his car around a turn on the track.*

*(f) A close up of a toy pony with blue hair.*

*(g) A piece of metal being melted in a furnace.*

Figure 9: **Additional Comparison on General Categories.**

