# OpenReview forum: "Towards Physical Understanding in Video Generation: A 3D Point Regularization Approach"
_NeurIPS.cc/2025/Conference — NeurIPS 2025 poster_

### Official Review · Reviewer_PwUi · 2025-06-24

**Clarity:** 2
**Significance:** 3
**Originality:** 3
**Rating:** 2
**Confidence:** 2

**Summary:**

This paper proposes injecting 3D point trajectories into video diffusion models to improve physical understanding. The authors introduce PointVid, a dataset with pixel-aligned 3D tracks, which is used to train models with point-based regularization to encourage smooth motion and shape consistency. The method improves video realism and can be applied to most existing diffusion models.

**Questions:**

1. Can the rigid loss be applied directly to object pixels in the RGB space instead of 3D point clouds? Would such an alternative achieve similar structure-preserving effects? An ablation would help clarify this.

2. Your framework relies on the first frame for segmentation and point tracking. How does the method generalize to longer or more complex videos with multiple moving objects or occlusions?

**Ethical Concerns:**

["NO or VERY MINOR ethics concerns only"]

**Final Justification:**

I thank the authors for their response and efforts. However, I still find the work insufficiently supported in terms of robustness and generalizability. The provided video results are only around 2 seconds in length, which is not adequate to demonstrate consistent performance in complex, multi-object, or long-term scenarios. Without experiments on longer sequences and corresponding evaluations, it is difficult to validate the method’s effectiveness beyond short clips. Therefore, I do not recommend acceptance.

**Limitations:**

Yes

**Quality:**

2

**Strengths And Weaknesses:**

Strengths：

1. The paper addresses an important limitation of current video diffusion models: lack of 3D physical understanding, through a simple yet effective solution.

2. The proposed framework is model-agnostic and can be integrated into most existing video diffusion architecture with minimal changes.

Weaknesses:

1. The rigid loss introduced for point clouds appears critical to preserving physical structure. However, it's unclear whether similar effects could be achieved by applying it directly in the RGB space. An ablation comparing point regularization to RGB-space alternatives would strengthen the paper.

2. The framework relies on the first frame for reference and sparse point tracking, which limits its scalability to long videos, multi-object scenes, or cases with occlusion and fast motion. The method is mainly demonstrated on short, single-object scenarios.

---

> ### Author Rebuttal · Authors · 2025-07-30
>
> Thank you for your constructive feedback. We appreciate your recognition of the contribution of our work. Below, we address your questions and concerns:
>
> **Q1 Can the rigid loss be applied directly to object pixels in the RGB space instead of 3D point clouds?**
>
> We propose the use of point trackings due to the lack of 3D spatial information in RGB space. Directly applying rigidity losses in RGB space is suboptimal. For instance, in a video of a race car moving from a distance toward the camera, the object’s size in the RGB frame increases from a small region of pixels to a large portion of the image. Applying rigidity loss directly to these pixel regions fails to preserve the object’s shape and may even cause undesirable shrinkage. To address this, we introduce 3D point tracking as a prior to overcome this limitation.
>
> Our ablation study in Figure 6(i) of the manuscript partially illustrates this issue, showing that training solely on RGB pixel differences does not achieve the desired 3D consistency. We agree that applying similar physical constraints directly in RGB space is a promising direction, provided that alternative methods can be developed to extract 3D information from RGB videos efficiently. We look forward to exploring such approaches in future work.
>
> **Q2 How does the method generalize to longer or more complex videos with multiple moving objects or occlusions?**
>
> The length of the generated video is constrained by the clip lengths in the training data. It can be generalized to predicting longer videos by scaling up the dataset. In our current training pipeline, we track foreground objects in the first frame and use point tracking to determine their 3D positions across time, accounting for possible interactions and occlusions between them. This added point modality enables our model to resolve multiple moving objects (see ‘Aston Martin’ example at 2’06’’ of the supplementary video) and handle contact-rich, multi-object scenarios with occlusion (see our task-oriented video examples). Even when an object is only partially visible or absent in the first frame, our model can still effectively handle it using the RGB modality. (For example, in the ‘tennis player’ case at 4’37’’, the player’s arm is not fully visible in the first frame, yet our model correctly infers its motion.)
>
> To extend our approach to longer videos, more complex scenes and object interactions, we plan to scale up the training data with richer scene compositions and more accurate, higher-resolution 3D tracking.

---

> > ### Comment · Reviewer_PwUi · 2025-08-04
> >
> > Thank you for your response and supplementary material. I have reviewed the provided videos, but most of them are short clips and lack systematic comparisons and quantitative evaluations in complex, multi-object, and long-term scenarios. As stated in Appendix A.2, the training videos are all cropped to approximately 2 seconds in length (“cropped to a 2-second length”), which further limits the evidence for the method’s applicability to longer temporal scales and more complex scenes. I suggest adding experiments on longer sequences (including multi-object, occlusion, and highly dynamic cases) and reporting corresponding quantitative metrics against baselines with stronger temporal modeling capabilities.

---

> > > ### Author Response · Authors · 2025-08-04
> > > **Reply to Official Comment by Reviewer PwUi**
> > >
> > > Thank you for your suggestions. We adopt different training resolutions for the videos based on what was used in the base model. In the UNet model, we used 16 frames; in the DiT model, we used 48 frames. We generated our dataset according to these clip lengths. We agree that experiments with longer video sequences would further strengthen the paper; however, this would require regenerating the dataset (as most video clips in the current dataset are under 60 frames) and retraining the model, which we are unable to accomplish within the rebuttal period. We look forward to upscaling the dataset as a future direction.

---

### Official Review · Reviewer_cz8h · 2025-07-02

**Clarity:** 4
**Significance:** 3
**Originality:** 3
**Rating:** 5
**Confidence:** 4

**Summary:**

The paper tackles the task of image-to-video generation, intending to inject explicit 3D understanding into diffusion-based video models. Starting from a pre-trained video generator, the authors propose to use a new modality: a three-channel map encoding of foreground objects' 3D points as pixel coordinates (u,v) and depth d. These point trajectories are obtained with an existing 3D point-tracking model, creating a dedicated training dataset PointVid. The diffusion model is then fine-tuned to predict both RGB frames and the (u,v,d) maps simultaneously. During the next training stage, the predicted point maps serve as an additional regulariser, guiding the network toward better 3D understanding and yielding stronger temporal and object consistency in the generated videos.

**Questions:**

1. Can you quantify the extra compute cost your regularizer adds compared to the standard diffusion loss? How does that overhead grow as you increase point density?
2. Have the authors conducted a systematic analysis of failure modes? In the supplementary video, the method exhibits improved temporal coherence but reduced motion amplitude. To what extent might the regularizer inadvertently suppress desirable motion dynamics, and how might this be mitigated?
3. Why choose a custom image subset over the public VBench split for evaluation? Could that decision skew your results or limit generality?
4. The technique seems readily applicable to text-to-video generation. Have you tested this scenario, and if so, can you share qualitative or quantitative results?
5. How does the number of tracked points affect output quality? Can you achieve similar results with a much sparser point cloud, and what trade-offs arise?

**Ethical Concerns:**

["NO or VERY MINOR ethics concerns only"]

**Final Justification:**

I remain positive about the paper and am keeping my original (accept) score, as the core idea is solid and the rebuttal clarified several technical points. However, two of my earlier concerns are still only partially addressed in the manuscript itself:

1. Evaluation split. The paper relies on a curated test subset. While the authors supplied numbers of other dataset in rebuttal, those results (and a short justification for using both splits) need to appear in the main paper so future work can compare directly.

2. Motion-regularisation trade-off. The supplementary video still shows reduced motion amplitude when the regulariser dominates. A brief analysis of this limitation, plus guidance on loss scheduling (e.g. how often the regulariser is applied, typical weight ratios), would help readers understand the quality-versus-motion balance and reproduce the method.

These points do not change my overall assessment, but incorporating them in the camera-ready would align the claims with the evidence and strengthen the final version.

**Limitations:**

Partially.
The manuscript lacks a failure-mode analysis, does not report the runtime or memory overhead introduced by the regulariser, and relies on a private evaluation set, preventing comparison with other improvement methods. Including these elements would substantially strengthen the paper.

**Paper Formatting Concerns:**

The manuscript’s formatting is satisfactory and requires no further revision.

**Quality:**

3

**Strengths And Weaknesses:**

Strengths:

1. Clarity of presentation. The paper is well-written, with clear explanations and informative visualisations that make the method easy to follow.

2. Novel 3D regularisation. Introducing an implicit 3D point-map modality and using it as a regulariser is a creative way to enhance geometric consistency in video diffusion models.

3. Compelling qualitative results. The method produces visually convincing videos that exhibit improved temporal and object consistency compared with baseline models.

Weaknesses:

1. Custom evaluation set. The main quantitative study uses a hand-curated subset of images, making it difficult to compare with existing or future work. Public benchmark suites such as VBench already support image-to-video evaluation and should be used for head-to-head comparisons.

2. Insufficient dataset analysis. The paper does not examine failure modes of the SpaTracker point-tracking stage or analyse tracking quality in the generated videos. A brief error study would strengthen the empirical section.

3. Missing cost breakdown. The additional regularisation loss incurs extra computation, but the paper does not report its runtime or memory overhead relative to the standard diffusion loss.

4. Limited evaluation scope. Results are presented only in the image-to-video scenario, even though the technique appears directly applicable to text-to-video generation. Demonstrating this extension would broaden the paper’s impact.

---

> ### Author Rebuttal · Authors · 2025-07-29
>
> Thank you for your constructive feedback. We appreciate your recognition of the novelty and effectiveness of our method. Below, we address your questions and concerns:
>
> **Q1 Can you quantify the extra compute cost your regularizer adds compared to the standard diffusion loss? How does that overhead grow as you increase point density?**
>
> We report the additional computational cost of our UNet model, trained on an A100 GPU with a batch size of 4 as follows: our 3D incorporation and regularization losses increase training times (∼9s per iteration, compared to ∼4.75s when fine-tuning on RGB video only) while maintaining comparable GPU memory usage (less than 2GB on top of ∼45GB). This additional computational cost is acceptable, as we are fine-tuning a pretrained video model rather than training one from scratch. Further statistics can be provided if needed.
> In our current design, the increased computational cost is primarily driven by the resolution of the training video rather than the density of the tracked points (see Q5 for more details).
>
> **Q2 To what extent might the regularizer inadvertently suppress desirable motion dynamics, and how might this be mitigated?**
>
> In our ablation studies (Figure 6 of the manuscript), we demonstrate that training with RGB alone lacks 3D awareness and results in failure cases, while using only the regularization loss leads to severe degradation of the RGB modality. In our approach, we adopt a joint optimization of diffusion and regularization losses, appropriately balancing them to be of comparable magnitude. This strategy yields a consistent decrease in regularization losses and a noticeable reduction in non-physical artifacts.
>
> We observe that when the regularization loss significantly outweighs the diffusion loss (by orders of magnitude), it tends to suppress motion. To address this, we apply the regularization loss less frequently, for example, once every five iterations. This adjustment helps preserve larger motion dynamics while requiring more training steps.
>
> **Q3 Why choose a custom image subset over the public VBench split for evaluation?**
>
> We use a custom test set for VBench to ensure consistency across all evaluation metrics, including generic video quality (using VBench), physical common sense (using VideoPhy), ablation studies on model design, and user studies (reported in the supplementary material). The test images are sampled from a single batch of video clips (unseen by our model) from video dataset [1], with the initial frame randomly selected within each clip.
>
> Below, we report the VBench results of our UNet-based model using their provided image set for your reference:
>
> | Item | SC | BC | MS | AQ | IQ |
> |----------|----------|----------|----------|----------|----------|
> | Baseline | 0.89703 | 0.94009 | 0.96993 |  0.52809 |  0.62961 |
> | Ours | 0.96418 | 0.96722 | 0.98319 | 0.51754 | 0.65428 |
>
> **Q4 The technique seems readily applicable to text-to-video generation. Have you tested this scenario?**
>
> We chose image-to-video tasks for a more controlled comparison of physical correctness, as the same objects are present in the scene. While our method is adaptable to text-to-video models, we did not evaluate them, as this is not our main focus. We look forward to exploring this direction in the future. As an alternative, text-to-video applications could be demonstrated by incorporating an additional text-to-image step prior to our model, and we are happy to include such examples if requested.
>
> **Q5 How does the number of tracked points affect output quality? Can you achieve similar results with a much sparser point cloud, and what trade-offs arise?**
>
> First, we would like to clarify that although we initially obtain point trajectories through sparse 3D tracking, during post-processing, we fill in all untracked foreground pixels (i.e., pixels inside the foreground mask) by interpolating from neighboring tracked points. Furthermore, they are stored as a tensor of the same dimension of the video and used for subsequent computations.  As a result, the point modality used in diffusion effectively becomes ‘dense’ tracking from the perspective of the video model. Thus, during training, the number of tracked points depends on how much of the reference frame is occupied by foreground objects, rather than on the tracking resolution used in data generation. On the other hand, during inference, we do not provide the point cloud, so there is no explicit control over point resolution.
>
> Currently, our dataset is generated using a fixed sparsity setting in SpaTracker [2]. We acknowledge that the tracking sparsity used during data generation affects accuracy of 3D information, and likely affects our training, we did not further explore this as this requires costly regeneration of large-scale dataset; we leave further investigation of this aspect for future work.
>
> **Q6 The paper does not examine failure modes of the SpaTracker point-tracking stage or analyse tracking quality in the generated videos.**
>
> Since we do not prompt the model with point tracking during the inference stage, we are unable to directly assess whether generation failures are caused by inaccurate tracking. To evaluate the quality of predicted point tracking during training, we report straightforward MSE loss measurements between the generated points and the ground truth:
>
> | Item | Untrained | Trained without Regularization | With Regularization|
> |----------|----------|----------|----------|
> | MSE | 0.21252 | 0.00310 | 0.00205 |
>
> [1] Chen T S, Siarohin A, Menapace W, et al. Panda-70m: Captioning 70m videos with multiple cross-modality teachers[C]//Proceedings of the IEEE/CVF Conference on Computer Vision and Pattern Recognition. 2024: 13320-13331.
>
> [2] Xiao Y, Wang Q, Zhang S, et al. Spatialtracker: Tracking any 2d pixels in 3d space[C]//Proceedings of the IEEE/CVF Conference on Computer Vision and Pattern Recognition. 2024: 20406-20417.

---

> > ### Comment · Reviewer_cz8h · 2025-08-04
> >
> > Thanks for the thoughtful rebuttal and the concrete numbers. I’m positive about the paper overall and will keep my score, but the camera-ready would benefit from integrating the key rebuttal details into the main text. In particular, please justify the choice of the curated evaluation subset and include the comparison on the official VBench split you reported, so readers can assess results on both a controlled set and a public benchmark. I’d also encourage a brief, candid discussion of the motion–regularization trade-off: the supplementary video still shows reduced motion amplitude, and I’m concerned that if the amplitude is preserved (e.g., by weakening or scheduling the regularizer), the current quality gains might diminish. An explicit analysis of this tension and implementation details on balancing the losses (for example, applying the regularizer every k steps and typical settings) would set expectations and make the method easier to reproduce.

---

### Official Review · Reviewer_Df51 · 2025-07-03

**Clarity:** 3
**Significance:** 2
**Originality:** 2
**Rating:** 5
**Confidence:** 4

**Summary:**

The paper proposes a video generation framework that incorporates 3D geometry and dynamic information into 2D videos. It introduces PointVid, a 3D-aware dataset created by augmenting videos with 3D point trajectories aligned in pixel space. By fine-tuning a latent diffusion model on this dataset, the method enables tracking of objects in 3D space and regularizes their shape and motion to reduce artifacts. This leads to improved visual quality and better handling of complex, contact-rich scenarios. The approach is compatible with existing video diffusion models to enhance their realism.

**Questions:**

Please check above weaknesses.

**Ethical Concerns:**

["NO or VERY MINOR ethics concerns only"]

**Final Justification:**

Most of the concerns I raised have been satisfactorily addressed during the rebuttal. After reviewing the feedback from other reviewers, I have decided to change my score from “borderline reject” to “accept.”

**Limitations:**

The authors addressed the limitations in main manuscript.

**Paper Formatting Concerns:**

No paper formatting concerns.

**Quality:**

2

**Strengths And Weaknesses:**

**Strength**

- The paper is clearly written and easy to follow. The authors effectively communicate their motivation and methodology in a straightforward manner. Additionally, the figures are clean and illustrative, contributing positively to the paper’s readability.

**Weaknesses**

- As acknowledged in the Limitations section, the proposed method inherits constraints from the base diffusion models. To better support the claimed generality of the plug-and-play design, it would be beneficial to include additional experiments with a broader set of base models. Using only I2VGen-XL and CogVideoX 1.5 is insufficient to convincingly demonstrate general effectiveness.

- The use of a VAE to encode latent features of the point modality raises concerns, as the VAE was not specifically trained on point-based data. It is unclear whether point regularization alone can adequately address this mismatch.

- I am concerned that the second and third terms in the reconstruction loss (Equation 1 of the main manuscript) may introduce a trade-off—such as improved temporal consistency at the cost of reduced temporal dynamics. It would be helpful if the authors could address this potential issue.

- The paper would benefit from a clearer description of the generated 3D trajectories. Are there any qualitative visualizations or quantitative metrics to evaluate the generated 3D points?

---

> ### Author Rebuttal · Authors · 2025-07-29
>
> Thank you for your constructive feedback. Below, we address your concerns:
>
> **Q1: Using only I2VGen-XL and CogVideoX 1.5 is insufficient to convincingly demonstrate general effectiveness.**
>
> We chose I2VGen-XL and CogVideoX 1.5 as representative models due to their distinct underlying architectures, which are based on UNet and Diffusion Transformer (DiT) frameworks, respectively. These two architectures are foundational, as most current video diffusion models share a similar backbone structure with one of them. We acknowledge that including experiments with additional baseline models would further strengthen our paper. However, due to constraints on computational resources, we leave this for future exploration.
>
> **Q2: VAE was not specifically trained on point-based data.**
>
> Using a VAE trained on RGB video to encode other modalities has been shown to be effective in prior work. For example, [1] uses a pretrained RGB VAE to encode depth maps without fine-tuning. Similarly, [2] leverages the RGB VAE to encode both depth and normal maps as input.
>
> During our training, we transform the point modality to the same range as color intensities ([-1, 1]) and reshape it to match the input format of the video. We observed that even without fine-tuning, the VAE is capable of reconstructing the main shape and motion of point clouds, while the point regularization further reduces noise (as shown in Figure 4 of our main manuscript). Additionally, we report that the mean squared error (MSE) of the encoded-decoded point tensor is less than 0.01, comparable to that of the reconstructed video tensor. Further statistics on point augmentation and regularization are provided in Q4.
>
> **Q3: The second and third terms in the reconstruction loss may introduce a trade-off.**
>
> In our design of the reconstruction loss in Equation 1, the first term (position) serves as the primary component for controlling macroscopic motion dynamics. The second and third terms (velocity and acceleration) act as complementary losses (with small weights) to suppress high-frequency, microscopic noise. While we acknowledge that incorporating motion smoothness losses introduces a trade-off, the global dynamics are primarily governed by the position term, and we observe faithful reconstruction of ground-truth dynamics during training. Moreover, discouraging abrupt changes in velocity is often beneficial, as such changes can lead to artifacts in the generated videos. For example, in the ‘race car’ case (5’00’’ in the supplementary video), highly dynamic baselines exhibit non-physical morphing, whereas our result demonstrates physically plausible motion while preserving a high degree of dynamism.
>
> **Q4: Evaluation of the generated 3D points**
>
> We report the MSE of predicted 3D point trajectories in diffusion (using the input point tracking tensor as ground truth) as a quantitative metric, summarized below:
>
> | Item | Untrained | Trained without Regularization | With Regularization |
> |----------|----------|----------|----------|
> | MSE | 0.21252 | 0.00310 | 0.00205 |
>
> We observe that our network design effectively predicts 3D point trajectories after the augmentation stage and further reduces the error through the regularization stage.
>
> [1] Shao J, Yang Y, Zhou H, et al. Learning temporally consistent video depth from video diffusion priors[C]//Proceedings of the Computer Vision and Pattern Recognition Conference. 2025: 22841-22852.
>
> [2] Liu X, Ren J, Siarohin A, et al. Hyperhuman: Hyper-realistic human generation with latent structural diffusion[J]. arXiv preprint arXiv:2310.08579, 2023.

---

> > ### Comment · Reviewer_Df51 · 2025-08-06
> > **Rebuttal Comment by Df51**
> >
> > Thank you for your thorough effort in addressing my concerns. I’m pleased to see that most issues and concerns have been resolved. I would like the points mentioned in the above rebuttal to be explained in detail in the final main script.

---

> ### Comment · Area_Chair_YxPh · 2025-08-06
>
> Dear Df51, we need you to show up and engage in the discussion phase.
> The guideline is:
> - read the author rebuttal
> - engage in discussions (reviewers must talk to authors, and optionally to other reviewers and AC - ask questions, listen to answers, and respond to authors)
> - fill in "Final Justification" text box and update “Rating” accordingly (this can be done upon convergence - reviewer must communicate with authors first)
>
> The (new) deadline is Aug 8, 11.59pm AoE

---

### Official Review · Reviewer_gf94 · 2025-07-03

**Clarity:** 3
**Significance:** 3
**Originality:** 3
**Rating:** 5
**Confidence:** 3

**Summary:**

This paper proposes a new video generation framework that integrates 3D geometries and motions. The key to the proposed method is: 2) The use of 3D point tracking as an additional modality along with RGB, 2) Regularization on the shapes and motions of objects represented by 3D points.

**Questions:**

Can the proposed method handle the cases where new objects appear in the mid of videos?

**Ethical Concerns:**

["NO or VERY MINOR ethics concerns only"]

**Limitations:**

Yes

**Quality:**

3

**Strengths And Weaknesses:**

Strength:
* The proposed framework with additional 3D point modality is well motivated, with simple and effective design.
* The proposed framework can be seamlessly integrated into existing video diffusion models for better physical plausibility.
* The paper is well-written and easy-to-follow.

---

> ### Author Rebuttal · Authors · 2025-07-29
>
> Thank you for your constructive feedback. We appreciate your recognition of the effectiveness of our method. Please find our response to your question below:
>
> **Q1: Can the proposed method handle the cases where new objects appear in the mid of videos?**
>
> Yes, our method can handle these cases effectively. For objects not present in the first frame, our model treats them as background and relies on the RGB modality to predict them. For example, in the ‘Aston Martin’ case (2’06’’ in the supplementary video), a second race car appears midway through the video, despite not being visible in the first frame. Since the RGB modality is jointly trained with the point modality, our method can accurately handle such scenarios. A similar situation occurs when objects are only partially visible initially but are still correctly modeled in the RGB space—for instance, the human hand in the ‘tool’ example at 3’42’’. We would be happy to include additional visual results in the revision if needed.

---

> ### Comment · Area_Chair_YxPh · 2025-08-06
>
> Dear gf94, I see your review is positive, and maybe nothing has changed, but we need you to show up and say something, and then click the "mandatory acknowldgement" button.

---

### Comment · Area_Chair_YxPh · 2025-08-04
**Discussion time**

Dear reviewers,

The authors appear to have put substantial work into their rebuttal. Is there anything you wish to discuss with them? Thank you PwUi for engaging already. As a reminder: author-reviewer discussion will be closed on August 6 11:59pm AoE.

Best,

Your AC

---

### Decision · Program_Chairs · 2025-09-17

**Decision:**

Accept (poster)

**Comment:**

This paper presents a method for video generation based on incorporating 3D point tracks as a prior. The idea is: given a pre-trained video generator, add 3 new channels to encode the foreground objects' 3D point coordinates (UVD), and then learn to predict the RGB together with the UVD map, and then also use the point maps as a space for applying regularizations. The overall effect is to reduce the well-known "morphing" artifacts of current video generators The paper also introduces a new pseudo-labelled dataset (corresponding to the zeroth step of the pipeline, where point tracks need to be pre-computed for some training videos).  This paper received mixed reviews: 5, 5, 5, 2. The reviewers largely find the approach well motivated, and the paper well-written. One review is confidently negative, pinpointing that the method is only demonstrated on two-second clips, which is perhaps too short to support claims about alleviating issues to do with shape consistency. The positive reviews contain their caveats as well, pointing out for example that the trade-offs of the motion regularizer (and details of how it should be applied in practice) are missing from the current draft, making it potentially difficult to reproduce the work. The AC also notes that while the paper heavily relies on the recent advancements in point tracking, and is one of many papers that are finding point trackers useful for video generation, the authors only cite the point tracker that they happen to use (SpaTracker); some advancements in point tracking, and video generative methods that rely on point trackers (e.g., Track4Gen, VideoJAM), should be acknowledged and ideally compared against, to better situate the work (and perhaps justify 3D point tracking instead of 2D as those works used). Overall, the AC sides with the majority positive reviews and recommends that the paper be accepted. The authors are encouraged to carefully revise the work to incorporate feedback gathered during the review process, in addition to the work already done during the rebuttal.